# Nutritional, Medicinal, and Cosmetic Value of Bioactive Compounds in Button Mushroom (*Agaricus bisporus*): A Review

**Muhammad Usman [1], Ghulam Murtaza [2] and Allah Ditta [3,4,*]**

[1] Department of Botany, Government College University Lahore, Lahore 54000, Pakistan; usmanphytologist@gmail.com
[2] Faculty of Environmental Science and Engineering, Kunming University of Science and Technology, Kunming 650500, China; murtazabotanist@gmail.com
[3] Department of Environmental Sciences, Shaheed Benazir Bhutto University Sheringal, Upper Dir, Khyber Pakhtunkhwa 18000, Pakistan
[4] School of Biological Sciences, The University of Western Australia, 35 Stirling Highway, Perth, WA 6009, Australia
[*] Correspondence: ad_abs@yahoo.com or allah.ditta@sbbu.edu.pk

**Abstract:** Fungi are vital to numerous industrial and household processes, especially producing cheeses, beer, wine, and bread, and they are accountable for breaking down organic matter. The remarkable medicinal and nutritional values of the mushrooms have increased their consumption. *Agaricus bisporus* belongs to the Agaricaceae family, and it is a top-ranked cultivated mushroom that is well known for its edibility. *A. bisporus* is rich in nutrients such as carbohydrates, amino acids, fats, and minerals and has potential anticancer, antioxidant, anti-obesity, and anti-inflammation properties. The bioactive compounds extracted from this mushroom can be used for the treatment of several common human diseases including cancer, bacterial and fungal infections, diabetes, heart disorder, and skin problems. *A. bisporus* has opened new horizons for the world to explore mushrooms as far as their culinary and medicinal values are concerned. In recent years, tyrosinase and ergothioneine have been extracted from this mushroom, which has made this mushroom worth considering more for nutritional and medicinal purposes. To emphasize various aspects of *A. bisporus*, a comprehensive review highlighting the nutritional, medicinal, and cosmetic values and finding out the research gaps is presented. In this way, it would be possible to improve the quality and quantity of bioactive compounds in *A. bisporus*, ultimately contributing to the discovery of new drugs and the responsible mechanisms. In the present review, we summarize the latest advancements regarding the nutritional, pharmaceutical, and cosmetic properties of *A. bisporus*. Moreover, research gaps with future research directions are also discussed.

**Keywords:** *Agaricus bisporus*; culinary; bioactive compounds; tyrosinase; ergothioneine

## 1. Introduction

With an increase in the world population, interest in the cultivation and subsequent consumption of mushrooms as a food source has increased. Since 1990, the world started focusing on the mushroom industry, and it resulted in a rapid increase in its production [1,2]. Mushrooms have become one of the most important sources of functional food and medicines in recent years [3,4]. The demand for edible mushrooms has increased due to their taste, flavor, and nutrient content [5,6]. Mushrooms are better alternatives to animal proteins and other animal products, and this fact has been supported through various studies conducted in the past [5,7–9]. Several forms of vitamins in mushrooms are responsible for improving health by decreasing the risk of various diseases in humans [10].

*Agaricus* is one of the largest genera of macrofungi, with several edible species that have medicinal and high nutritional values [11]. *Agaricus bisporus* (J. E. Lange) Imbach, a

member of the Agaricaceae family, ranks at the top among cultivated mushrooms, and is well known for its edibility. It is considered as one of the most important mushrooms based on its culinary and medicinal values [12]. Production and consumption of this mushroom have been consistently increasing for the last six to seven decades. China ranks at the top with the highest production of *A. bisporus*. It exports *A. bisporus* mainly to Russia, Japan, Vietnam, Korea, and Thailand and in low amounts to Australia, as well as several European and African countries [13].

Button mushroom is another name of *A. bisporus* and it is a valuable source of food and several important bioactive compounds [14]. Several important bioactive compounds have been isolated from *A. bisporus* during the past few years [15]. The bioactive compounds with nutritional value in *A. bisporus* contribute to human health. Several studies have also reported the role of this mushroom in the cosmetics industry as it contains certain constituents that enhance facial beauty by controlling various skin problems [8,16,17]. A summary of the nutritional, medicinal, and cosmetic value of *A. bisporus* is given in Figure 1.

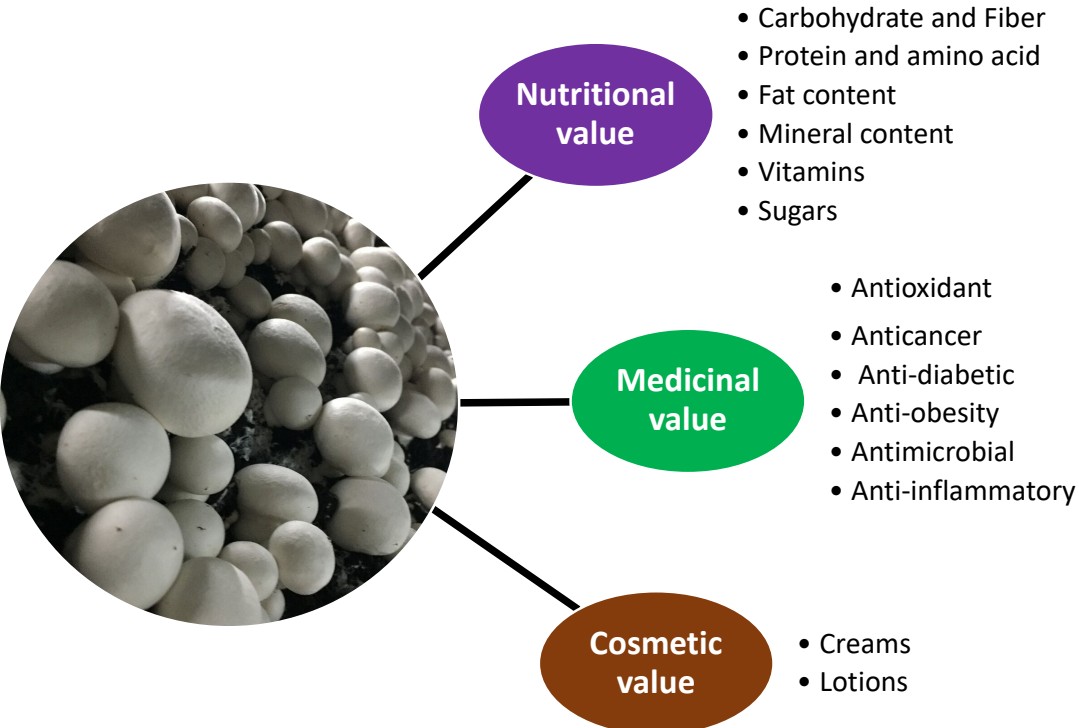

**Figure 1.** A summary of nutritional, medicinal, and cosmetic value of *Agaricus bisporus*.

From Figure 1, it is clear that *A. bisporus* has great potential from nutritional, medicinal, and cosmetic aspects. There is a need to emphasize these aspects, and this could only be possible through a comprehensive review of multidimensional aspects of *A. bisporus* and finding out the research gaps. In this way, the quality and quantity of bioactive compounds in *A. bisporus* would be enhanced, ultimately contributing towards nutritional, medicinal, and cosmetic aspects. The new research work may also result in the discovery of new drugs from the bioactive compounds in *A. bisporus* by employing latest state-of-the-art technologies. Based on these hypotheses, the present review aims to have a comprehensive view of the nutritional, medicinal, and cosmetic value of bioactive compounds in *Agaricus bisporus*.

## 2. Nutritional Value

Various studies have been conducted to evaluate the nutritional value of *A. bisporus* [8,18–25].

Tables 1 and 2 represent the various nutritional components of *A. bisporus* found in different studies. *A. bisporus* is relatively rich in carbohydrates, proteins, and fats as compared to some other widely consumed species [26]. According to El Sebaaly et al. [27], mushrooms exhibit different nutritional value when grown on different substrates.

**Table 1.** Nutritional status of *A. bisporus*.

| Raw Nutritional Value/100 g | |
|---|---|
| Energy | 94 KJ (22 kcal) |
| Water | 92.43 g |
| Fats | 0.34 g |
| Proteins | 3.09 g |
| Carbohydrates | 3.26 g |
| Dietary fibers | 1 g |
| Sugar | 1.65 g |
| Iron | 0.50 mg |
| Vitamin C | 2.1 mg |
| Niacin (Vit B3) | 3.607 mg |
| Riboflavin (Vit B2) | 0.402 mg |
| Pantothenic acid (B5) | 1.497 mg |

Source: USDA Nutrient database.

**Table 2.** Nutritional components present in the basidiocarp of *A. bisporus*.

| Nutritional Group | Components | Amount | Unit | References |
|---|---|---|---|---|
| Proteins (amino acids) | Alanine | 5.8 | g 100 g$^{-1}$ total protein in FW | [28–30] |
| | Cysteine | 1.1 | | |
| | Proline | 6.1 | | |
| | Tyrosine | 4.2 | | |
| | Methionine | 0.8 | mg g$^{-1}$ DW | |
| | Threonine | 1.3 | | |
| | Asparagine acid | 3.4 | | |
| | Serine | 3.1 | | |
| | Histidine | 14.1 | | |
| | Leucine | 0.8 | | |
| | Arginine | 2.2 | | |
| | Lysine | 3.5 | | |
| | Phenylalanine | 2.1 | | |
| | Glycine | 2.0 | | |
| | Valine | 2.3 | | |
| | Isoleucine | 1.0 | | |
| | Total amount of amino acids | 44.2 | | |
| Lipids (fatty acids) | Palmitic acid | 13.35 | mg 100 g$^{-1}$ DW | [31] |
| | Palmitoleic acid | 4.84 | | |
| | Caprylic acid | 1.08 | | |
| | Caprinic acid | 0.85 | | |
| | Oleic acid | 6.07 | | |
| | Linoleic acid | 67.29 | | |
| | Linolenic acid | 1.52 | | |
| | Laurnic acid | 0.11 | | |
| | Myristic acid | 0.94 | | |
| | Stearic acid | 3.72 | | |
| | Arachidic acid | 0.92 | | |
| | Pentadecanoic acid | 0.23 | | |
| | Total unsaturated fatty acids | 79.72 | | |
| | Total saturated fatty acids | 20.28 | | |
| | Total lipids | 2.7 | % DW | |

**Table 2.** *Cont.*

| Nutritional Group | Components | Amount | Unit | References |
|---|---|---|---|---|
| Carbohydrates | Total sugars | 4.50 | g 100 g$^{-1}$ FW | [30,32] |
| | Fructose | 2.62 | | |
| | Mannitol | 23.62 | | |
| | Trehalose | 1–3 | % DW | |
| Indol compounds | Indoloacetic acid | 0.19 | mg 100 g$^{-1}$ DW | [8,22] |
| | Tryptamin | 0.06 | | |
| | Kynurenic acid | 6.21 | | |
| | Melatonin | 0.11 | | |
| | Serotonin | 5.21 | | |
| | L–Tryptophan | 0.39 | | |
| Vitamins | Niacin | 42.0 | mg 100 g$^{-1}$ DW | [28,33,34] |
| | Vitamin B1 | 0.6 | | |
| | Vitamin B2 | 5.1 | | |
| | Vitamin B3 | 43.0 | | |
| | Vitamin C | 17.0 | | |
| | γ-Tocopherol | 2–3 | | |
| | α-Tocopherol | 1–4 | | |
| | δ-Tocopherol | 1.0 | | |
| | Folic acid | 450 | μg 100 g$^{-1}$ | |
| | Vitamin B12 | 0.8 | | |
| | Vitamin D | 3.0 | | |
| Phenolic compounds | Ferulic acid | 42.83 | mg kg$^{-1}$ DW | [23,30,35,36] |
| | Gallic acid | 280.45 | | |
| | Cinnamic acid | 0.38 | | |
| | Myricetin | 2729.46 | | |
| | Caffeic acid | 392.51 | | |
| | Catechins | 56.74 | | |
| | Procatechuic acid | 83.26 | | |
| | *p*-Coumaric acid | 2.31 | | |
| | Total phenols | 277–687 | | |
| | Free phenols | 176–487 | | |
| Sterols | Ergosta-7, 22-deiniol | 2.45 | mg 100 g$^{-1}$ DW | [37,38] |
| | Ergosterol | 186.1 | | |
| | Ergosta-7-enol | 1.73 | | |
| | Ergosta-5,7-deiniol | 6.05 | | |

Note: mg = Milligram; kg = Kilogram; g = Gram; DW = Dry weight; FW = Fresh weight.

### 2.1. Proteins and Amino Acids

Mushrooms contain relatively high protein contents compared to animal products, but usually rank below animal meats [8]. However, it is reported that pre- and post-harvest conditions affect the nutritional and chemical composition of mushrooms. Moreover, different mushrooms' developmental stages exhibit different amounts of protein and amino acid contents [20,26,39,40]. *A. bisporus* is rich in different forms of amino acid (Figure 2).

The amino acids continuously assimilate to produce urea, which contributes to the overall mushroom's total nitrogen contents [41]. Post-harvested mushrooms were reported to shown protease activity [42]. Mushrooms growing on different substrates display variable protein contents as reported by different researchers around the world. Braaksma and Schaap [43] reported that major forms of amino acids in *A. bisporus* include aspartic acid, serine, glycine, threonine, glutamine, valine, cysteine, alanine, leucine, isoleucine, lysine, histidine, proline, arginine, tyrosine, and norleucine. Moreover, crude protein contents in *A. bisporus* ranged from 19–38% on a dry weight basis. Several studies in the past have focused on determining the overall protein and amino acid content of *A. bisporus*. According to Sadiq et al. [44], 11.01% protein contents were found in *A. bisporus*. Muszynska et al. [22] recorded the presence of

proteins up to 11.01% in *A. bisporus*. However, Mohiuddin et al. [45] found that *A. bisporus* contained 17.7–24.7% protein contents while Ahlavat et al. [25] reported the presence of 29.1% protein contents in *A. bisporus*. These results indicate that the protein contents vary in *A. bisporus* based on the usage of different growth substrates.

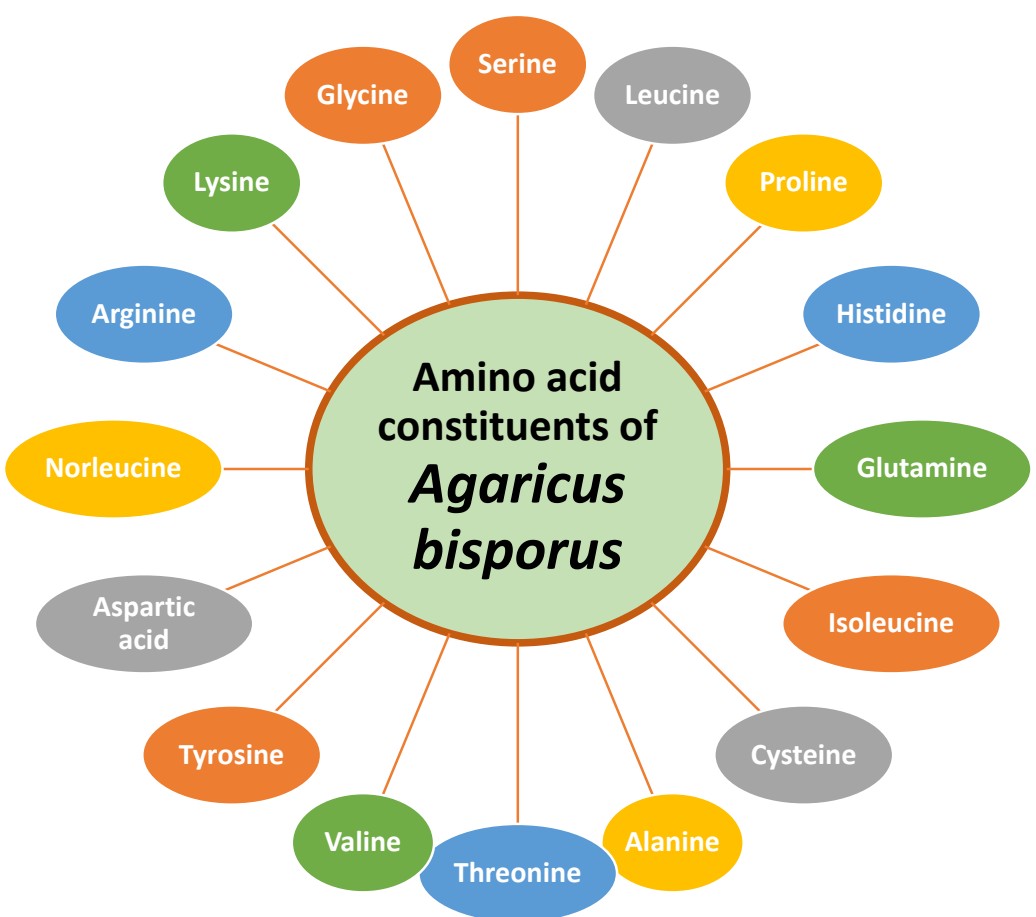

**Figure 2.** Amino acid constituents in *Agaricus bisporus* [43].

Recently, a novel class of specific ribonucleases, known as "ribotoxin-like proteins," have been discovered [46,47]. It has been found that ribotoxins are widely distributed among various species of mushrooms. Citores et al. [46] isolated a ribotoxin, i.e., "Ageritin," from *Agrocybe aegerita* and found that it possessed multiple biological activities, i.e., antibacterial, antiviral, endonuclease, nuclease, and cytotoxic activities, which could be employed in plants through transgenic techniques to enhance resistance against viruses, bacteria, and fungi.

### 2.2. Lipids

Crude fats are present only in small amounts while a considerable amount of essential fatty acids like linoleic acid are present in *A. bisporus*. As compared to *A. bisporus* cultivated strains, wild *Agaricus* sp. contains relatively higher amounts of polyunsaturated fatty acids and a lower concentration of monounsaturated fatty acids [19]. According to Baars et al. [48], the total fatty acid contents ranged from 180–5818 mg kg$^{-1}$ DW and linoleic acid represented almost 90% of the fat contents present in *A. bisporus*.

The fatty acid constituents in *A. bisporus* were stearic acid, palmitic, linoleic, caprylic, oleic, erucic, and eicosanoic acid, which accounted for about 44.19% of the total extracted fatty acid [44]. Two fatty acids, i.e., palmitic (12.67–14.71%) and linoleic acids (61.82–67.29%), are the main fatty acids in *A. bisporus* among the detected 13 fatty acids [31]. However, Shao et al. [21] reported palmitic, linoleic, and stearic acids as the major fatty

acids found in *A. bisporus*. Hossain et al. [49] investigated the principal fatty acids in two important edible mushrooms, i.e., *Pleurotus ostreatus* and *Ganoderma lucidum* (Curtis) P. Karst., and found that linoleic acid was five-fold and twenty-fold more than the total fatty acids measured, respectively.

### 2.3. Carbohydrate and Fibers

Undoubtedly, carbohydrates are a basic component of human food, and mushrooms are a significant source of both forms of carbohydrates, i.e., digestible and non-digestible. Glucose and mannitol are the two important digestible carbohydrates but are present in low quantities (not more than 1% of the dry weight) in *A. bisporus*. Moreover, glycogen is another digestible carbohydrate, which ranges from 5 to 10% of the dry weight in *A. bisporus*. Trehalose, mannans, and β–glucan are non-digestible carbohydrates, which constitute a major portion of the carbohydrate contents in *A. bisporus* [8,50]. Reis et al. [23] examined the carbohydrate contents in *A. bisporus* and found that trehalose and mannitol were the two most abundant sugars. Mannitol represents the most plentiful sugar in *A. bisporus* [48]. Mannitol was the principal form of soluble sugar in fresh fruiting bodies of this species, followed by glucose ranging from 17.6–28.1 mg g$^{-1}$ in several phases of its maturity [51,52].

Cheung [50] stated that the dietary fibers included chitin, which is predominated in the fungal cell wall. Mannans, hemicelluloses, and glucans are also present in significant amounts in the fungal cell wall, which, as a result, contribute to the medicinal properties of the mushrooms. Atila et al. [53] determined the chitin contents in *A. bisporus* as 9.60 g 100 g$^{-1}$ dry matter. Vetter [54] found that the chitin contents were much higher in *A. bisporus* compared to *Lentinula edodes* (Berk.) Pegler and *P. ostreatus* (Jacq. ex Fr.) P.Kumm. It is important to understand that chitin is an insoluble fiber, which promotes immune functioning and improves gut health. Higher chitin contents also contribute towards *A. bisporus* health. Cherno et al. [55] measured the chitin contents in *A. bisporus* and concluded that it had approximately two times more chitin contents in comparison to *P. ostreatus*.

### 2.4. Minerals

*A. bisporus* is considered an important source of minerals, predominately rich in copper (Cu), cobalt (Co), iron (Fe), selenium (Se), potassium (K), and manganese (Mn) [20,52]. The principal mineral constituents of fruiting bodies in mushrooms are phosphorus (P) and potassium (K), followed by calcium (Ca), zinc (Zn), iron (Fe), magnesium (Mg), and sodium (Na). These minerals have been found in the *A. bisporus* and impart their role in the health benefits of the mushroom [56].

Mohiuddin et al. [45] investigated the mineral contents in *A. bisporus* growing under different locations. The mineral contents (mg kg$^{-1}$) ranged in the samples as 37.2–61.9 for sodium (Na), 54.6–163.4 for copper, 56.2–91.1 for magnesium (Mg), 143.6–396 for iron (Fe), and 36.6–58.0 for zinc (Zn). Caglarırmak [57] also determined different mineral contents in *A. bisporus* as calcium (534.2–554.8 mg kg$^{-1}$), potassium (213.3–238.8 mg kg$^{-1}$), Fe (7.4–7.9 mg kg$^{-1}$), Zn (8.1–8.7 mg kg$^{-1}$), Na (2652–2500 mg kg$^{-1}$), Mg (88.0–76.3 mg kg$^{-1}$) and phosphorus (7.4–7.9 mg kg$^{-1}$). Similarly, Ahlavat et al. [25] calculated the mineral content in the fruiting bodies of *A. bisporus* as Na (500.8 mg kg$^{-1}$) and Se (1.34 mg kg$^{-1}$). Lu and Holmgren [58] stated that Se in *A. bisporus* is a crucial micronutrient for animals and humans.

### 2.5. Vitamins

Various studies have confirmed that mushrooms are a valuable source of vitamins. Bernas & Jaworska [59] stated that the most abundant vitamins in *A. bisporus* included niacin and riboflavin. Other important vitamins included α-tocopherol, ascorbic acid, vitamin B1, and vitamin B3. Caglarırmak [57] stated that the *A. bisporus* is rich in vitamins such as riboflavin, folic acid, thiamin, and niacin, but poor in vitamin C content. Furlani and Godoy [60] calculated the mean values of vitamins B1 (Thiamin) and B2 (Riboflavin) from a fresh specimen of *A. bisporus* and found that vitamin B1 and B2 were 0.03 and

0.25 mg 100 g$^{-1}$, respectively. Moreover, vitamin B2 amounts in *A. bisporus*, *Pleurotus* spp,. and *L. edodes*, except for conserved mushrooms, were greater than that of various vegetables. Ahlavat et al. [25] demonstrated that this species is an ample source of vitamin D (984 IU g$^{-1}$). Simon et al. [61] stated that vitamin D contents vary in wild and cultivated strains of *A. bisporus*. Reis et al. [23] found that the poor contents of vitamin D could be due to the cultivation of species in darkness. Roberts et al. [62] reported that exposure to UV radiation helps to enhance the production of vitamin D. Ergo-sterol is also present in the fungal cell wall and is a precursor of vitamin D2. Shao et al. [21] stated that ergo-sterol content usually correlates with the mushroom's antioxidant activity.

## 3. Medicinal Value

With time, there has been an increased interest in consuming mushrooms to cure or treat many deadly diseases worldwide. *A. bisporus* is the best example of mushrooms that has great medicinal and nutritional properties. *A. bisporus* has long been used in traditional therapies in many different states and countries. Various studies have confirmed that bioactive compounds, powder, and even extract from *A. bisporus* can be used to treat various deadly human diseases. This trend is increasing with every passing day. It has potential anticancer, antioxidant, anti-obesity, and anti-inflammation properties and could be used to treat coronary heart diseases, cancer, diabetes mellitus, disorders related to the immune system, viral, bacterial and fungal infections [30,63]. Hence, the consumption of *A. bisporus* makes the human body more resistant to various diseases as it boosts the immune functioning of the body [64]. However, comparatively few studies have reported the direct treatment trials of mushrooms using humans to confirm the medicinal potential of *A. bisporus* against various diseases [65]. Various studies have confirmed anticancer activity [53,66–69], antioxidant [19,70,71], and anti-diabetic [72–77] properties of *A. bisporus*. The following sections give a detailed description of each medicinal property of *A. bisporus*. Recently, a novel class of specific ribonucleases, known as "ribotoxin-like proteins", have been discovered [46,47]. It has been found that ribotoxins are widely distributed among various species of mushrooms and are responsible for various biological activities against bacteria, viruses, and fungi.

### 3.1. Antioxidant Properties

*A. bisporus* exhibits comparatively higher antioxidant potential compared with other important edible mushrooms like *P. eryngii* (DC.) Quel., *Grifola frondosa* (Dicks.) Grey, *P. ostreatus,* and *L. edodes* [14,53]. Oms-Oliu et al. [70] reported 100.32–100.78 mg 100 g$^{-1}$ phenolic content based on the fresh weight in fresh-cut *A. bisporus*. Moreover, ergothioneine contents from brown and white *A. bisporus* ranged from 0.21–45 mg g$^{-1}$ DW. Liu et al. [71] examined the chief phenolic compounds from ethanolic extract of *A. bisporus* and confirmed the presence of natural antioxidants, such as catechin, ferulic acid, gallic acid, protocatechuic acid, and myricetin. Some other researchers have also confirmed the presence of antioxidants in *A. bisporus* [53,78–81].

Sarikaya & Gulcin [82] stated that serotonin is another important biochemical compound that exhibits great antioxidant potential. Because of its antioxidant potential, serotonin extracted from mushrooms have a potential to prevent Alzheimer's disease [83]. Muszynska et al. [22] calculated the serotonin content from fruiting bodies of *A. bisporus*, which was 5.21 mg 100 g$^{-1}$ dry weight. Tocopherol is a class of fat-soluble vitamins and an important antioxidant involved in many body functions. White *A. bisporus* was found rich in β-tocopherol, i.e., 0.85 μg 100 g$^{-1}$ fresh weight [23].

### 3.2. Anticancer Properties

Over the last few decades, cancer has remained one of the few life-threatening diseases on Earth. Recent reports have confirmed that the polysaccharides extracted from some mushrooms inhibit cancer cell lines because of their significant anticancer activity [84]. Polysaccharides from *A. bisporus* have significant action against cancerous cells via en-

hancing cellular immunity. According to the Canadian Cancer Society, increased use of *A. bisporus* enhances immune functions as it exhibits effectiveness against some deadly diseases [85]. *A. bisporus* contains a large amount of several important polysaccharides, which have anti-tumor and strong immune-stimulatory activity in vitro and in vivo [86].

Smiderle et al. [87] reported three principal polysaccharides in *A. bisporus*, namely galactomannan, α–glucan, and β–glucan, contributing about 55.8%. McCleary and Draga [88] reported that the low β-contents in this species are beneficial in enhancing mucosal immunity. Jeong et al. [68] stated that a significant dietary fiber level speeds up the secretion of immunoglobulin A.

Jagadish et al. [89] reported that lectins and phenolic compounds in *A. bisporus* suppress tumorous cells. The study was carried to show the effect of bioactive compounds extracted from *A. bisporus* on cancer line cells. The results showed that proliferation of HL-60 leukemic cells was inhibited by *A. bisporus* extract as it stimulated programmed cell death [90]. Similarly, *A. bisporus* extract suppressed the growth of prostate cancer PC3 and DU145 cells of mice [91,92].

*A. bisporus* extract containing lectins was effective against lung cancer cells by strengthening the cellular mechanism of antioxidant defense and inhibiting the growth of cancerous cells [68]. Arginine from basidiocarp of *A. bisporus* retards the growth of tumor cells. The study recommended that persons particularly cancer patients should use this particular mushroom regularly as dietary supplements. ABP-1 and ABP-2 are polysaccharide fractions that suppressed the growth of breast cancer cells (MCF-7), but their effect on gastric cancer, prostate cancer, sarcoma, and colorectal carcinoma was not statistically significant [67,93].

Selenium (Se) is an essential trace element for animals, including humans. According to Clark et al. [94], Se contents are found more in mushrooms compared to the fruits and vegetables. The *A. bisporus* is a valuable source of trace minerals that contributes to its health benefits. Se has a crucial role in the chemoprevention of cancer. Spolar et al. [95] reported that the Se makes the immune system stronger to fight against some deadly diseases including cancer. Rzymski et al. [96] stated that the higher doses of Se significantly alleviate the chances of cancer. Major types of cancer including lung, liver, colon, and prostate can be prevented by the antioxidants present in *A. bisporus*.

### 3.3. Anti-Diabetic Activity

Various studies have confirmed the anti-diabetic activities of *A. bisporus* [30,65,73–76, 97–100]. *A. bisporus* holds significant levels of antioxidants such as vitamin C, D, and B12, as well as polyphenols, folates, and dietary fibers, which have inhibitory effects against diabetes and cardiovascular disorders [75]. Calvo et al. [76] isolated a variety of bioactive compounds from *A. bisporus* that have the potential to defend the body against type 2 diabetes. Ekowati et al. [100] found that *A. bisporus* extract significantly increased insulin production and G6PD activity with a substantial decrease in glucose concentration in the rat's body. Similarly, the use of *A. bisporus* also caused a significant increase in low-density lipoprotein (LDL) levels, cholesterol, and triglyceride. Moreover, a decrease in malondialdehyde level was substantial while superoxide dismutase, catalase, and glutathione peroxidase were significantly increased [100].

The high amount of dietary fibers (about 19%) and some other valuable carbohydrate constituents in *A. bisporus* may result in its glucose-lowering effect. The effect of dietary fibers from *A. bisporus* is similar as observed in other high-valued mushrooms [97,101]. The lectin-like molecules in *A. bisporus* are responsible for decreasing glucose and increasing insulin contents. *A. bisporus* is contributing to lower blood glucose levels due to high fiber contents. High fiber contents in *A. bisporus* act as a barrier against the action of digestive enzymes, which lowers the blood glucose level [73,74].

Propionate, an ester of propionic acid extracted from *A. bisporus,* contributes to its anti-diabetic properties [102,103]. Propionate is associated with gluconeogenesis and regulation of serum lipid levels and is reported to lower blood glucose levels. Wong et al. [104]

investigated the possible mechanisms in lowering the body's blood glucose level using *A. bisporus*. It was found that fibers in *A. bisporus* are involved in bacterial fermentation, which contains an ample amount of oligosaccharide and polysaccharide and ultimately results in increased production of short-chain fatty acids in the colon.

Volman et al. [65] demonstrated that the intake of α–glucan lowered the lipopolysaccharide production, which resulted in a substantial decrease in glucose concentration. Ekowati et al. [100] reported that terpenoids and flavonoids are secondary compounds that increase SOD activity in rats suffering from diabetes. Huang et al. [98] reported that flavonoids increased the expression levels of endogenous antioxidant genes and consequently increase SOD activity. Terpenoids are involved in the modulation of SOD activity [99]. The high contents of zinc (7.5–15 mg 100 g$^{-1}$ DW) in the basidiocarp of *A. bisporus* significantly improve the SOD activity [30].

### 3.4. Anti-Obesity Activity

Hyperlipidemia is a dominant risk factor represented by raised levels of triglyceride or cholesterol. It results in the most serious heart diseases and atherosclerosis in humans [105]. Mushrooms are highly nutritive, exhibiting tremendous amounts of bioactive compounds (alkaloids, flavonoids, polysaccharides, polyphenols, fibers, sterols, and terpenes) that have antioxidant potential with positive effects on numerous cardiac biomarkers in treating obesity and obesity-related cardiovascular illnesses [81]. Lin et al. [106] reported that phytosterols or plant sterols reduce cholesterol absorption. Phytosterols extracted from the *A. bisporus* reduce LDL cholesterol and plasma cholesterol. Xu et al. [107] stated that *A. bisporus* also exhibits a substantial amount of lovastatin, which lowers the cholesterol levels in the body to reduce the chances of cardiovascular disorders.

Jeong et al. [75] demonstrated that the consumption of *A. bisporus* basidiocarp regulates anti-cholesterolemic and anti-glycemic responses in rats fed with a high cholesterol diet and rats suffering from type 2 diabetes induced by streptozotocin injection (50 mg kg$^{-1}$ body weight). The results showed that the *A. bisporus* has both anti-hypercholesterolemic and anti-glycemic effects in rats. Additionally, it imparted a positive effect on lipid metabolism as well as on liver functioning.

### 3.5. Antimicrobial Activity

*A. bisporus* contains many bioactive compounds responsible for antimicrobial activities. The use of traditional synthetic medicines in providing relief against plants, animals, and human diseases has not proven quite productive [108]. Research has examined that gram-positive bacterial growth and development were greatly inhibited via methanol extract from wild *A. bisporus*. Methanol extract had little impact on the growth and development of gram-negative bacteria in comparison to gram-positive bacteria [31]. Mycochemical analysis and activities of *A. bisporus* are presented in Table 3.

**Table 3.** Mycochemical analysis and activities in *Agaricus bisporus*.

| Mycochemicals | Activities | References |
|---|---|---|
| Alkaloids | Antimicrobial, Anti-inflammatory, Antioxidant | [53,109] |
| Carbohydrate | Antimicrobial | [20,31,53,110,111] |
| Phenols and Polyphenols | Antimicrobial, Anti-inflammatory, Antioxidant | [53,109,112] |
| Protein and Amino acids | Antimicrobial, Anti-inflammatory | [113] |
| Saponins | Anticancer, Antioxidant | [114] |
| Tannins | Antimicrobial, Antioxidant | [115] |

Shang et al. [116] reported similar findings from the Chinese strain, where *A. bisporus* extract did not affect gram-negative bacteria. On the other hand, Soltanian et al. [117] examined the effect of crude extract on the growth and development of gram-negative and gram-positive bacteria. In the results, crude extract from both cultivated and wild *A. bisporus* only showed inhibitory effects on gram-positive bacteria. Contrary to this, there

are also reports about the repressive effect of *A. bisporus* crude extract on gram-negative bacteria, and the main susceptible species included *Listeria monocytogenes*, *Escherichia coli*, *Klebsiella pneumonia*, and *Pseudomonas aeruginosa* [118–120].

In the secondary metabolites of *A. bisporus* extract, a few compounds have antimicrobial activity. Only a few studies have been conducted to investigate which particular compound is responsible for antimicrobial activity. However, during the last few years, some valuable findings have been published [121]. The aqueous protein extract from cultivated *A. bisporus* has shown a significant antibacterial activity, specifically against *Staphylococcus aureus* [122].

The growth of two bacterial species, namely *Escherichia coli* and *S. aureus,* was significantly inhibited via ethanol and cold-water extract from *A. bisporus* [123]. Delgado-Povedano et al. [124] found that the potential antiviral activity of aqueous enzymatic extracts of *A. bisporus* could prevent HCV infection due to the antagonistic activity of the extract against HCV protease.

### 3.6. Anti-Inflammatory Properties

Some studies have confirmed the anti-inflammatory properties of *A. bisporus* [15,93,125, 126]. Mannogalactan, fucomannogalactan, and fucogalactan are the polysaccharides isolated from *A. bisporus* that exhibit analgesic and anti-inflammatory properties [126]. Heterogalactan is another polysaccharide extracted from this species that can combat sepsis in mice [125]. Golak-Siwulska et al. [93] stated that sepsis is a very grievous medical problem and one of the principal causes of death in intensive care units throughout the world, and heterogalactan from the *A. bisporus* can be helpful in these cases. Ruthes et al. [15] demonstrated that *A. bisporus* could fight sepsis as it exhibits a strong anti-inflammatory effect.

## 4. Cosmetic Value

The mushrooms have been mainly investigated for their medicinal and nutritional values [127]. However, little work has been done in exploring the biotechnological potential of mushroom extract for the formulation of plenty of cosmetic products [128,129]. It has been found that skin whitening could be achieved by inhibiting the tyrosinase enzyme activity [130–132]. The cosmetic industry has been working continuously to isolate some important ingredients from mushrooms and confirm their aesthetic values and their subsequent use in cosmetic products such as lotions and creams [133].

Intrinsic or natural mechanisms lead to skin aging. This process considerably affects the skin and various other body parts through hormonal alterations happening with age and exposure to ultraviolet radiations, which result in the generation of reactive oxygen species (ROS), causing oxidative stress [134]. Activator factor-1 (AP-1) is one important transcription factor accelerated by ROS and is known to enhance elastin and carries out the breakdown of collagen by up-regulation of Matric Metallo-Proteinase [135]. In the modern world, people are more interested in using natural compounds for skincare. In addition, the demand for natural compounds is increasing due to their defensive and protective role against free radical generation and reduction in oxidative enzyme production. Tyrosinase enzyme extracted from the *A. bisporus* has been found effective against inflammation and inflammatory diseases [136].

Ethanolic extract from *A. bisporus* exhibits strong antioxidant properties, thereby playing an important role in curing skin aging. Extracts from *A. bisporus* possess antibacterial activity against harmful microorganisms including methicillin-resistant and methicillin-susceptible *Staphylococcus aureus*. These bacteria are known to colonize the skin during injury and inflammation. The base cosmetic cream prepared from mushroom extracts displayed anti-inflammatory and antioxidant potential against the production of nitric oxide and melanin by suppressing the tyrosinase activity [137]. Cosmetic formulations comprising mushroom extracts suppress the growth of bacterial strains, which cause damage to the skin. Certain bioactive compounds in the base cream extracted from mush-

rooms enhance beauty and play an important role against inflammation, skin aging, and hyper-pigmentation [133].

The base cream prepared from the *A. bisporus* extract displayed a significant antioxidant potential and antibacterial action against some bacterial strains [120,129]. The extract from this species containing a bioactive compound (2-Amino-3H-phenoxazin-3-one) utilized in specific concentrations, e.g., at 0.5, 1, and 2 μM, inhibited the production of melanin by 80, 54.1, and 39.4%, respectively [137]. Phenolic compounds possess antibacterial activity that usually disrupts the bacterial membranes and damages DNA [138]. Alves et al. [139] found that the *A. bisporus* extract showed a non-significant resistance against two gram-negative bacteria, i.e., *E. coli* and *P. aeruginosa*.

*A. bisporus* is an edible mushroom that often accumulates Se as a trace mineral in their fruiting bodies [140,141]. Shampoos can be formulated using *A. bisporus* extract in resolving hair problems like dandruff, oily hair, and hair loss. It can also be used for cleansing the hair and scalp for better growth of the hairs [142]. *A. bisporus* contains vitamin D and several minerals including copper, iron, and Se, which play an important role in making hairs healthy and strong, thereby preventing hair loss and dandruff [127]. The trend of using mushrooms in the cosmetic industry should increase with time.

## 5. Conclusions and Future Perspectives

*Agaricus bisporus* is not only a valuable source of food but also exhibits medicinal and cosmetic values. During the last two decades, interest in the consumption of *A. bisporus* has increased, predominantly in developed and developing countries as supplements to healthy food. Many studies have confirmed that the bioactive compounds isolated from *A. bisporus* are promising for drugs against some deadly diseases. Therefore, an increasing trend in the use of *A. bisporus* has been observed due to its robust medicinal properties like anticancer, antioxidant, anti-diabetic, antimicrobial, and anti-obesity activity. Recently, tyrosinase and ergothioneine have been reported from this mushroom, making this mushroom more worth considering for nutritional and medicinal purposes. However, there is room for researchers to explore the cosmeceutical properties of this mushroom. In addition, more data are required to evaluate and confirm the exact mechanisms behind the treatment of specific diseases and to discover some novel drugs in this regard.

**Author Contributions:** Writing—original draft preparation, M.U. and G.M.; writing—review and editing, A.D.; supervision, A.D. All authors have read and agreed to the published version of the manuscript.

**Funding:** This research received no external funding.

**Institutional Review Board Statement:** Not applicable.

**Informed Consent Statement:** Not applicable.

**Acknowledgments:** The authors are thankful to Muhammad Ehsan, (visiting faculty of English literature at the University of Narowal, Narowal) for helping in improving the grammar.

**Conflicts of Interest:** The authors declare no conflict of interest.

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
