# Peer review of "Nutritional, Medicinal, and Cosmetic Value of Bioactive Compounds in Button Mushroom (Agaricus bisporus): A Review"

_applsci, doi:10.3390/app11135943_

Round 1

Reviewer 1 Report

Your research was adequately addressed.

Author Response

Response to the reviewer comment

Comment: Your research was adequately addressed.

Response: Thanks for your recommendation

Reviewer 2 Report

For the authors:

Lines 25–27. “To emphasize …”. This is not a complete sentence. Perhaps, after “the research gaps”, “is presented” should be added.

Lines 29–31. This is a matter of style, but I prefer the present tense here, viz. “we summarize” rather than “we have summarized”, and “are also discussed” rather than “have also been discussed”.

Line 36. Delete “been”, i.e. just use “has increased”.

Line 39. Delete “been”.

Line 81. I think “on different substrates” is better than “under different substrates”.

Line 82, Table 2. Arginine is listed twice. Also, myristic acid is spelled incorrectly.

Line 86. “rank” rather than “ranked” and delete “the”. That is, it should read “rank below animal meats”.

Lines 93–94. “display” rather than “displayed”.

Line 125. Should this be “a basic component” rather than “the basic component”? It is not the only component.

Line 142. “ex” rather than “Ex”.

Line 173. “vegetables” rather than “veggies”.

Lines 190–191. It should read “the consumption of A. bisporus makes the human body more resistant …”

Line 192. Replace “fewer studies have reported the” by “few studies that have reported”.

Line 200. Italics needed for species name.

Line 202. No capital E needed for “ergothioneine”.

Line 217. Replace “earth” by “Earth” or “this planet”.

Line 249. Replace “lungs” by “lung”.

Line 252. Delete “the” before the species name.

Line 315. Replace “In contrary” by “In contrast to this” or “Contrary to this”.

Lines 317–318 and lines 323–325. Italics needed for species names.

Line 336. Replace “drives to” by “causes of”.

Line 362. Italics needed for species name.

Line 366. Delete “of” after “comprising”.

Line 368. Delete “the” before “beauty” and replace “plant” by “play”.

Line 377. Italics needed for species name.

Line 379. Replace “bodies” by “fruit bodies”.

Line 388. Delete “been”.

Line 396. Replace “data is” by “data are”.

Line 405. “literature” shouldn’t have a hyphen.

Author Response

Response to the reviewer 2 comments

For the authors:

Comment: Lines 25–27. “To emphasize …”. This is not a complete sentence. Perhaps, after “the research gaps”, “is presented” should be added.

Response: Corrected. (Please see line 26)

Comment: Lines 29–31. This is a matter of style, but I prefer the present tense here, viz. “we summarize” rather than “we have summarized”, and “are also discussed” rather than “have also been discussed”.

Response: Corrected. (Please see line 29)

Comment: Line 36. Delete “been”, i.e. just use “has increased”.

Response: Corrected. (Please see line 31)

Comment: Line 39. Delete “been”.

Response: Corrected. (Please see line 36)

Comment: Line 81. I think “on different substrates” is better than “under different substrates”.

Response: Corrected. (Please see line 85)

Comment: Line 82, Table 2. Arginine is listed twice. Also, myristic acid is spelled incorrectly.

Response: Corrected. both. (Please see Table 2)

Comment: Line 86. “rank” rather than “ranked” and delete “the”. That is, it should read “rank below animal meats”.

Response: Corrected. (Please see line 104)

Comment: Lines 93–94. “display” rather than “displayed”.

Response: Corrected. (Please see line 110-111)

Comment: Line 125. Should this be “a basic component” rather than “the basic component”? It is not the only component.

Response: Corrected. (Please see line 150)

Comment: Line 142. “ex” rather than “Ex”.

Response: Corrected. (Please see line 167)

Comment: Line 173. “vegetables” rather than “veggies”.

Response: Corrected. (Please see line 272-273)

Comment: Lines 190–191. It should read “the consumption of A. bisporus makes the human body more resistant …”

Response: Corrected. (Please see line 215-216)

Comment: Line 192. Replace “fewer studies have reported the” by “few studies that have reported”.

Response: Corrected. (Please see line 217)

Comment: Line 200. Italics needed for species name.

Response: Corrected. (Please see line 228)

Comment: Line 202. No capital E needed for “ergothioneine”.

Response: Corrected. (Please see line 230)

Comment: Line 217. Replace “earth” by “Earth” or “this planet”.

Response: Corrected. (Please see line 245)

Comment: Line 249. Replace “lungs” by “lung”.

Response: Corrected. (Please see line 277)

Comment: Line 252. Delete “the” before the species name.

Response: Corrected. (Please see line 280)

Comment: Line 315. Replace “In contrary” by “In contrast to this” or “Contrary to this”.

Response: Corrected. (Please see line 349)

Comment: Lines 317–318 and lines 323–325. Italics needed for species names.

Response: Corrected. (Please see line 351-352 and 357-358)

Comment: Line 336. Replace “drives to” by “causes of”.

Response: Corrected. (Please see line 370)

Comment: Line 362. Italics needed for species name.

Response: Corrected. (Please see line 396)

Comment: Line 366. Delete “of” after “comprising”.

Response: Corrected. (Please see line 399-400)

Comment: Line 368. Delete “the” before “beauty” and replace “plant” by “play”.

Response: Corrected. (Please see line 401-402)

Comment: Line 377. Italics needed for species name.

Response: Corrected. (Please see line 411)

Comment: Line 379. Replace “bodies” by “fruit bodies”.

Response: Corrected. (Please see line 413)

Comment: Line 388. Delete “been”.

Response: Corrected. (Please see line 422)

Comment: Line 396. Replace “data is” by “data are”.

Response: Corrected. (Please see line 430)

Comment: Line 405. “literature” shouldn’t have a hyphen.

Response: Corrected. (Please see line 439)

Reviewer 3 Report

The review article entitled “Nutritional, medicinal, and cosmetic value of bioactive com-2pounds in button mushroom (Agaricus bisporus): A review” by Muhammad et al., reports the nutritional, medicinal, and cosmetic value of bioactive compounds in Agaricus bisporus.

Studying bioactive compounds from mushroom is a research field of great interest, especially in the last few years. However, there some major and minor changes to correct, as reported below:

Major changes:

Authors claim that edible mushrooms are characterized by several properties such as anticancer, anti-diabetic, anti-obesity, antimicrobial, anti-inflammatory properties and so on, due to the presence of some important bioactive compounds such as phenolic compounds, tocopherols, and ascorbic acid. This is right; however, they forget to mention that also proteins/enzymes are responsible for these activities. In particular, in the last few years, in literature is reported a novel class of specific ribonucleases, known as “ribotoxin-like proteins” (prototype Ageritin) retrieved in edible basidiomycete mushrooms that have been well characterized.

See for example:

- Citores et al. 2019 Jun 21;14(6):1319-1327

- Fogarasi M et al., Agronomy 2020, 10(12):1972

On the basis of these considerations, authors must revise the manuscript adding more information, considering these discoveries of the last few years in the field of edible mushrooms bioactive proteins.

Authors affirm that ‘’Mushrooms are the better alternatives of animal proteins and other animal products, and this fact is supported through various studies conducted in the past’’ In literature, there are many studies on these for example: Nagy, M. et al., Food Science and Technology (Campinas) 37(2) 315-320, Qing Z. et al., Lwt 135 (2021) 110063, Please add a reference.

Authors affirm that ‘The  demand  for  edible  mushrooms  has  been increased due to their taste, flavor, and nutrient content’ without reference to confirm this. In literature, there are many studies on these, for example:

  • Nagy M et al., Bulletin UASVM Food Science and Technology 74(1) / 2017
  • Fogarasi M et al., Molecules 23(12)

Minor changes:

Page 9, Line 283: change ‘dry weight’ to DW

Page 10, Line. 317, 323, 325: Please write with Italic the name of the bacteria

Author Response

Response to the reviewer 3 comments

Comment: The review article entitled “Nutritional, medicinal, and cosmetic value of bioactive com-2pounds in button mushroom (Agaricus bisporus): A review” by Muhammad et al., reports the nutritional, medicinal, and cosmetic value of bioactive compounds in Agaricus bisporus. Studying bioactive compounds from mushroom is a research field of great interest, especially in the last few years. However, there some major and minor changes to correct, as reported below:

Response: Thanks for your appreciation. We have incorporated all the suggestions made by the reviewer

Major changes:

Comment: Authors claim that edible mushrooms are characterized by several properties such as anticancer, anti-diabetic, anti-obesity, antimicrobial, anti-inflammatory properties and so on, due to the presence of some important bioactive compounds such as phenolic compounds, tocopherols, and ascorbic acid. This is right; however, they forget to mention that also proteins/enzymes are responsible for these activities. In particular, in the last few years, in literature is reported a novel class of specific ribonucleases, known as “ribotoxin-like proteins” (prototype Ageritin) retrieved in edible basidiomycete mushrooms that have been well characterized. See for example:

- Citores et al. 2019 Jun 21;14(6):1319-1327

- Fogarasi M et al., Agronomy 2020, 10(12):1972

On the basis of these considerations, authors must revise the manuscript adding more information, considering these discoveries of the last few years in the field of edible mushrooms bioactive proteins.

Response: We have consulted the mentioned papers and added the required information regarding ribotoxin proteins (Please lines 126-132)

Comment: Authors affirm that ‘’Mushrooms are the better alternatives of animal proteins and other animal products, and this fact is supported through various studies conducted in the past’’ In literature, there are many studies on these for example: Nagy, M. et al., Food Science and Technology (Campinas) 37(2) 315-320, Qing Z. et al., Lwt 135 (2021) 110063, Please add a reference.

Response: We have consulted the mentioned papers, cited and in the reference list (Please see line 42)

Comment: Authors affirm that ‘The  demand  for  edible  mushrooms  has  been increased due to their taste, flavor, and nutrient content’ without reference to confirm this. In literature, there are many studies on these, for example:

Nagy M et al., Bulletin UASVM Food Science and Technology 74(1) / 2017

Fogarasi M et al., Molecules 23(12)

Response: We have consulted the mentioned papers, cited and in the reference list (Please see line 40)

Minor changes:

Page 9, Line 283: change ‘dry weight’ to DW

Response: Corrected. (Please see line 311)

Comment: Page 10, Line. 317, 323, 325: Please write with Italic the name of the bacteria

Response: Corrected. (Please see line 351-352)

Round 2

Reviewer 2 Report

I am providing no further comments.

Reviewer 3 Report

After the careful revision done by authors, the article is now acceptable for publication on Applied Sciences journal.